# Learning to Navigate Wikipedia
# by Taking Random Walks

**Manzil Zaheer**\*, **Kenneth Marino**\*, **Will Grathwohl**\*, **John Schultz**\*, **Wendy Shang**,
**Sheila Babayan, Arun Ahuja, Ishita Dasgupta, Christine Kaeser-Chen, Rob Fergus**
DeepMind New York
{manzilzaheer, kmarino, wgrathwohl, jhtschultz, wendyshang,
sbabayan, arahuja, idg, christinech, robfergus}@google.com

## Abstract

A fundamental ability of an intelligent web-based agent is seeking out and acquiring new information. Internet search engines reliably find the correct vicinity but the top results may be a few links away from the desired target. A complementary approach is navigation via hyperlinks, employing a policy that comprehends local content and selects a link that moves it closer to the target. In this paper, we show that behavioral cloning of randomly sampled trajectories is sufficient to learn an effective link selection policy. We demonstrate the approach on a graph version of Wikipedia with 38M nodes and 387M edges. The model is able to efficiently navigate between nodes 5 and 20 steps apart $96\%$ and $92\%$ of the time, respectively. We then use the resulting embeddings and policy in downstream fact verification and question answering tasks where, in combination with basic TF-IDF search and ranking methods, they are competitive results to the state-of-the-art methods.

## 1 Introduction

The ability to gather new knowledge about the world is a fundamental aspect of intelligence. Based only on a few words, a fact, question, or even vague idea, humans have the ability to use the internet to find extremely specific information about the world. From the important (how to administer CPR) to the trivial (the LA Raiders won the 1983 SuperBowl), virtually any knowledge is a click away.

In this work, we focus in one particular aspect of this ability: web navigation. In general, navigation is a key component of an embodied agent: the ability to move efficiently toward a target. In known environments, where map information is available, shortest-path algorithms provide a viable solution. However, in novel settings, the agent lacks such global information and must instead *navigate* to the target, using its understanding of the local environment to select actions.

Other approaches to web agents have focused mostly on retrieval or search engines. These, however, only provide a partial solution, typically getting close to a desired target but often not exactly to the right page. In this work, we present an approach for navigation on graph-structured web data that uses hyperlinks within articles to navigate toward a target. It complements search engines, using them to provide a sensible starting point for a local search for specific information.

To be able to navigate on the web effectively, we must read the text of the page, see how concepts in the current page are related to a possible next page, and use our understanding to learn a policy to make the right navigation decision. Consider the scenario shown in Figure 1. The agent is trying to navigate to the target node NORTH AMERICA while currently at the node PRESIDENTS OF THE UNITED STATES containing two hyperlinks: BARACK OBAMA and USA. Our approach learns an

---

\*Denotes equal contribution

36th Conference on Neural Information Processing Systems (NeurIPS 2022).

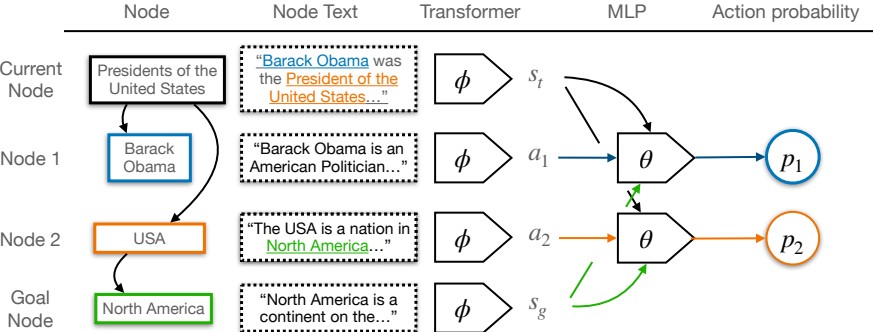

Figure 1: Our agent navigates through hyperlinks on a Wikipedia graph towards a target node. It first embeds the text at each link node using Transformer $\phi(.)$ and then evaluates different actions/links $a_1, a_2$ with a policy $\theta$, conditioned on the current and target embeddings $s_t$ and $s_g$, respectively.

embedding from the text of the current page and the text for each hyperlink node and passes these to a goal-conditioned policy that selects the best possible action. To make the correct choice in this example, the agent should associate USA with NORTH AMERICA.

The particular web environment we consider here is Wikipedia, converted into a graph form. Each of the 38M paragraphs is represented by a node; edges are links within and between articles. We introduce an approach for navigation on this graph based on behavioral cloning of automatically generated trajectories. This allows for unsupervised pre-training of both web page embeddings and navigation policy. We explore different trajectory distributions for training the model, the best practical choice being equivalent to random walks on the graph.

**Contributions:** Our work presents the first viable solution to the problem of web navigation[1], as demonstrated by reliable (>90% success) navigation on a graph of size $10^7$. Previous attempts such as Nogueira and Cho [2016] show a far inferior performance, well below that needed for practical utility. Furthermore, as there is nothing Wikipedia-specific about our approach and it scales gracefully, the method is applicable to general web settings. We also demonstrate the relevance of the pre-train + fine-tune paradigm to the web domain, showing that pre-training for navigation is a suitable objective to aid downstream tasks with limited data such as Q&A. We demonstrate this by applying our navigation approach on two challenging tasks: fact verification (FEVER; Thorne et al. [2018]) and Q&A (Natural Questions; Lee et al. [2019]) for evidence gathering. In both settings, we show a jump in performance over existing search-based approaches that rely on retrieval followed by re-ranking, matching state-of-the-art methods despite using far simpler retrieval and re-ranking mechanisms.

## 2  Background

**Wikipedia as a navigation environment:** Human behavior on Wikipedia [West and Leskovec, 2012a] was collected via the Wikispeedia game [West et al., 2009], which asked the players to move between a pair of arbitrarily chosen articles in the fewest steps. This rule allows the emergence of a semantic distance between concepts that facilitates retrieval. West and Leskovec [2012b] conducted initial studies on this dataset for navigation, using supervised learning and reinforcement learning methods on a small subset of articles (4.6K) with simple TF-IDF features.

Most relevantly, Nogueira and Cho [2016] explore the same problem of navigation on a large Wikipedia graph (12M nodes). They train LSTM agents to perform navigation on the graph, using Bag-of-Word features for each article. The trained agent is then applied to a Jeopardy Q&A task. Our approach revisits this framing but with a number of important differences: (i) training on a significantly larger graph (38M vs 12M nodes, 51.5M vs 380M edges); (ii) using Transformer-based architectures for paragraph encodings and for the navigation policy; (iii) they use a single fixed start node, whereas ours is able to navigate from any start node; (iv) removal of beam search, or other complex search mechanisms at evaluation time. Collectively, our approach boosts the navigation success rate from $12.5\%$ in Nogueira and Cho [2016] to over $90\%$,[2] despite a much larger graph.

---

[1]A demo of our agent navigating is shown here `https://www.youtube.com/watch?v=LVqOaPKpC2c`

[2]Exact comparisons are infeasible given idiosyncrasies in graph construction and other differences in setup.

**Fact verification and Q&A using Wikipedia:** Wikipedia is a rich resource for evaluating language understanding through Q&A tasks. Chen et al. [2017] combine TF-IDF retrieval with an LSTM document reader to find answers to free-form questions within Wikipedia documents. Contemporary Q&A systems [Semnani and Pandey, 2020, Qu et al., 2020] extend this general approach to more complex pipelines, combining traditional and neural methods with inverted index search/retrieval schemes, followed by document readers that perform re-ranking. We demonstrate our navigation approach in conjunction with very basic retrieval and re-ranking approaches, so as to clearly demonstrate the value of navigation. In common with our approach, Wang et al. [2021] also use a graph version of Wikipedia, but with the goal of aligning it to a knowledge base, rather than navigation. Stammbach and Neumann [2019] propose a scheme for finding evidence for the FEVER [Thorne et al., 2018] benchmark that explores all links one step away from an initial retrieval set. This can be viewed as a 1-step navigation operation, in contrast to our scheme that permits an arbitrary number of steps.

**Navigation in knowledge bases:** Graph navigation is a key element in utilizing knowledge bases (KB), where facts are represented as tuples of entity nodes connected by a labelled edge corresponding to relation between the two nodes. For example, Das et al. [2017] answer complex queries by navigating the graph, inferring missing edges along the way, and finding a valid path to the answer node. In our case, the graph is fixed but consists of natural language text and is partially observed. This makes the task much harder on two fronts. First, the policy must comprehend the natural language actions, without access to crisply labelled relation actions. Second, the action space is open and varying in size, unlike in a KB which typically has a fixed schema, i.e. relations/actions come from a fixed size set. Perhaps the closest work involving natural language actions, Fu et al. [2019] augment a KB with textual nodes. However, it still operates under the fixed KB schema. Further improvements on navigation in KB have been shown by incorporating tricks like reward-shaping [Xiong et al., 2017] and action dropout [Lin et al., 2018]. We adopted the action dropout where some outgoing edges are masked during training to enable more effective path exploration. Pushing more on exploration, inspired by AlphaGo, Shen et al. [2018], He et al. [2022] combine the policy with Monte-Carlo tree search to navigate KB. We find that no search method is necessary for our approach at evaluation time, with good performance obtained from the policy alone.

**Internet powered Q&A:** Several recent works have attempted complex question answering using the internet. Talmor and Berant [2018] retrieve snippets of information from the web with a search engine to answer a structured decomposition of the question with a simpler Q&A model. The recent WebGPT [Nakano et al., 2021] system takes a broadly similar approach to long-form question answering. WebGPT accures information from the Bing search API as well as some navigation, and passes it to an LM to generate the answer text. The complexity of the system necessitates the use of human demonstrations for training. Related work from Lazaridou et al. [2022] utilizes a few-shot supervision of an LM to perform Google queries, which are then used by the LM to generate an answer. They show that by incorporating information from Google queries, the answers are more factual than those generated directly from LMs. Another line of work, has leveraged link structure of webpages found on the internet to improve Q&A. In LinkBERT [Yasunaga et al., 2022] is pretrained to capture dependencies between documents and results in improvement for for multi-hop reasoning and few-shot reading comprehension. Asai et al. [2020] use the link structure to enhance open domain Q&A by learning to retrieve according to reasoning paths. In contrast to these works, we focus on mastering navigation rather than any combination with search/retrieval, albeit in more limited domain (Wikipedia vs the internet). However, our approach could potentially act as a core component in a general web navigation, being more versatile on account of not requiring supervision.

# 3 Approach

We consider navigation in a web environment $\mathcal{W} = \{\mathbf{N}, \mathbf{L}\}$, where $\mathbf{N} := \{n_i\}_{i=1}^{N}$ is the set of nodes, each $n_i$ representing a web page with content $s_i$. The links between pages are represented by $\mathbf{L} := \{l_i\}_{i=1}^{N}$, where $l_i = \{l_i^1, \ldots, l_i^j, \ldots, l_i^{M_i}\}$ are the set of $M_i$ (directed) out-links from node $i$. We define $\mathrm{NH}(n_i)$ as the set of nodes reached by following all links $l_i$ from node $n_i$, where the $j$th neighbor is given by $l_i^j(n_i)$.

We assume some target node $n_g$ and some starting node $n_0$ are given, and wish to train an agent that can navigate from $n_0$ to $n_g$ by walking along the edges of the graph within $T$ steps. To accomplish this, we construct a parametric policy $p_\theta(n_{t+1}|n_t, n_g)$ which, at time step $t$, parameterizes a distribution over $n_{t+1} \in \mathrm{NH}(n_t)$. We train this model via behavioral cloning to maximize

$$\mathcal{L}(\theta) = \mathbf{E}_{\mathcal{E}(n_0,\dots,n_T)} \left[ \sum_{t=0}^{T-1} \log p_\theta(n_{t+1}|n_t, n_g) \right] \qquad (1)$$

where $\mathcal{E}(n_0,\dots,n_T)$ is our trajectory distribution which generates trajectories $(n_0,\dots,n_T)$ that we would like our model to replicate (i.e. setting $n_g = n_T$).

## 3.1 Trajectory distribution

This framework opens up many choices for $\mathcal{E}$. Clearly some will result in ineffective navigation strategies while others may enable us to train a policy which is capable of generalizing to new graphs and downstream applications. We elaborate on some choices below.

**Reverse trajectories:** We define $\mathcal{E}(n_T) = \text{Uniform}(\mathbf{N})$ (the uniform distribution over nodes in the graph). Now we can define

$$\mathcal{E}(n_t|n_{t+1}) = \frac{1}{|\{n_i; n_i \in \text{NH}(n_t)\}|} \quad \text{if } n_{t+1} \in \text{NH}(n_t), \quad \text{else } 0 \qquad (2)$$

or, a distribution which uniformly walks the *reverse* graph (with transposed adjacency matrix). We begin with some uniformly chosen target node and walk backward randomly to $n_0$ in such a way that we can reach the target by walking forward from $n_0$.

Intuitively, training with the reverse trajectories defined above should allow our model to be able to reach any node in the graph from any node it is connected to. Unfortunately, in realistic web graph data, there exist many "dead-ends" or nodes with no in-going edges. The trajectories get stuck at these nodes leading to low-diversity training data.

**Random forward trajectories:** In our real-world knowledge graph data, there are far fewer nodes which have no out-going edges. Thus we propose an alternative method to generate target nodes based on randomly walking forward. We define $\mathcal{E}(n_0)$ to be the $\text{Uniform}(\mathbf{N})$ and

$$\mathcal{E}(n_{t+1}|n_t) = \frac{1}{|\text{NH}(n_t)|} \quad \text{if } n_{t+1} \in \text{NH}(n_t), \quad \text{else } 0. \qquad (3)$$

or, a distribution which uniformly picks a starting node and randomly walks the *forward* graph for $T$ steps. Training in this way should encourage our model to be able to reach any node which is reachable from some uniformly chosen start. We note that the target node distribution $\mathcal{E}(n_T)$ will not be uniform and instead be a function of the graph structure. We also note that while this generates a $T$-step trajectory, because it is generated by random walk, the shortest path distance between start and target nodes may be smaller.

**Random shortest paths:** We define $\mathcal{E}(n_T) = \text{Uniform}(\mathbf{N})$ and let $\mathcal{E}(n_0|n_T)$ be the uniform distribution of nodes whose shortest path length to $n_T$ is $T$ steps. We then define $\mathcal{E}(n_t|n_0, n_T)$ to be the delta distribution that $n_t$ is the $t$-th node of this shortest path. While it can be challenging to compute the probabilities of this distribution efficiently we can draw samples from it using a shortest path finding algorithm such as Dijkstra's algorithm.

## 3.2 Model parameterization

From above, we can see the core of our model is the $p_\theta(n_{t+1}|n_t, n_T)$ distribution. Given a sampled trajectory that begins at $n_0$, this is the distribution which attempts to choose the correct action to take at timestep $t + 1$ starting at node $n_t$ in order to get to target node $n_T$. We parameterize this distribution as a function of the text available at node $n_t$.

$$p_\theta(n_{t+1}|n_t, n_g) \propto \exp(f_\theta(s_{t+1}, s_t, s_g, t)) \qquad (4)$$

where $f_\theta$ embeds the text $s_g$, $s_t$ and $s_{t+1}$ of the target, current and destination nodes, respectively. Further details on how we apply this general model to Wikipedia are given in Section 4.2 and an alternative interpretation of this approach based on variational inference in a latent-variable model is presented in Appendix D.

# 4 Application to Wikipedia

## 4.1 Navigation and sentence search tasks

**Data processing:** We convert a snapshot of English Wikipedia into a graph but split each article ($\bar{\mu} = 1000$ words) into paragraph-sized blocks ($\bar{\mu} = 100$ words), which form the *nodes*. *Edges* in the graph are the union of (i) organic hyperlinks, (ii) additional entity linking and (iii) next/previous paragraph links. We consider two snapshots from year 2017 and 2018, which produces large full graphs with $\sim$ 37M nodes and $\sim$ 370M edges. For initial experiments, we also use a smaller subsampled graph, for which we sub-sample disjoint sets of 200k train / 200k evaluation nodes from the 2018 graph. All the obtained graphs are well connected with median path length of 15. Further details have been relegated to the Appendix (full graphs in Appendix A.1, the 200k graph in Appendix A.2, and graph statistics in Appendix A.3).

**T-step navigation:** The most basic task is $T$-step navigation. At the start of the episode, we randomly sample a start node and generate a $T$-step random walk trajectory. We then take the last node of this trajectory as the target $n_g$. The agent is then given a time budget $B$ (in most experiments, we fix max steps to $B = 100$), and succeeds if it reaches the target node within $B$ steps. In our experiments, we show results on various graphs and splits with $T = \{5, 10, 20\}$-step navigation.

**1-T Multistep navigation:** The Multistep task is much like the previous setting, except that instead of generating fixed $T$-step trajectories, we sample from Uniform$(1, T)$, $T = 20$.

**T-step sentence search:** This is the same setup as $T$-step navigation, except that rather than provide the full text for the target node $n_g$, only one sentence within that text is selected at random to compute the target embeddings from. Thus agents have the more challenging task of locating the correct node, given only a small snippet of its text (see Section 4.2 for details on target representation). We include this task because one of the difficulties of web navigation is establishing which page should be the target in the first place. It also has direct relevance to our downstream fact verification task.

## 4.2 Model architecture

**State representation:** For each node $n_i$ we embed its text $s_i$ using a Transformer $\phi(.)$ to produce $\phi(s_i)$. By basing the state representation on semantic content, rather than i.e. node index, the agent is able to learn about the relationships between entities in a manner that allows for generalization to new graphs at evaluation time, instead of memorizing the structure of the training graph. Following Karpukhin et al. [2020], we append to the node text the title of the Wikipedia article to which it belongs. This often provides context to the node that might otherwise be missing – in many articles, the subject of the article will be referred to indirectly (e.g. it might state that "she" was born in 1945 instead of repeating the subject's name).

For those experiments where the node embeddings are kept fixed, $\phi(.)$ is a pre-trained RoBERTA [Liu et al., 2019] model that encodes the node text & title. However, in our large-scale experiments, we pre-train the Transformer $\phi(.)$ directly using the text & title (see Appendix B.1). In both cases, we apply the Transformer to the tokenized text, take the mean over the input tokens, and apply a `tanh` nonlinearity to produce $\phi(s_i)$.

**Target representation:** Similarly, we use the article text $s_g$ of the target node as input to the Transformer model $\phi$ to compute its representation $\phi(s_g)$. For some tasks, such as sentence search or fact verification, the target representation instead relies on a single sentence from each node or other text specific to the task and a separate transformer model $\phi_{\text{target}}$ is trained to embed the target.

**Navigation policy network:** The network operates on the target node $n_g$ and the current node $n_t$, and outputs a distribution over the possible actions that can be taken at $n_t$. As the nodes in the graph have a variable number of outgoing edges, the policy must produce a distribution with a variable number of outcomes. To do this, we first concatenate the embeddings of $s_g$ and $s_t$, and pass them through a 1-layer feed-forward network to produce a combined embedding $e_{tg} = FF[\phi(s_t), \phi(s_g)]$. Next we embed each possible action $a_i$. When we use a fixed $\phi$, we directly use $a_i = \phi(s_i)$ for $n_i \in \text{NH}(n_t)$. However this is not feasible when pre-training our embedding model due to memory concerns. Instead, we compute $a_i = \phi(s_t[l_t^i])$ where $s[l^i]$ selects the words in $s$ which belong to hyperlink $i$ – see Appendix B.1 for more details. We then define the probability of moving to $n_i$ from

$n_t$ with target $n_g$ as $p(n_i|n_t, n_g) \propto \exp\left(e_{tg} \cdot a_i\right)$ which is normalized to produce a distribution over the $|\text{NH}(n_t)|$ actions.

In our larger-scale model, we also include the trajectory leading up to the current node $(n_0, \ldots, n_t)$. We pass their embeddings, combined with the target embedding, $[\phi(s_0), \ldots, \phi(s_t), \phi(s_g)]$, through a 4-layer transformer and use the output as $e_{tg}$ when computing the transition probabilities above. We explore these two policy architecture choices, feed-forward and Transformer in Table 4.

## 4.3 Downstream tasks

Finally, we use our navigation approach to gather information for the tasks of fact verification on the FEVER benchmark [Thorne et al., 2018] and question answering on Natural Questions (NQ) [Lee et al., 2019] which also use Wikipedia as a knowledge base. One important difference from our earlier navigation task is that the target node $n_g$ is not specified at evaluation time. Instead, we are given a claim whose veracity must be established or a question that must be answered by finding information in a particular node in the graph. To determine which node this is, we use a target encoder which maps the claim or question to an embedding vector that can be used in place of $\phi(s_g)$ in Section 4.2.

Due to the limited size of the downstream training sets, we pre-train our graph embeddings, navigation policy and target encoder on the similar sentence search task. We then freeze the embeddings and policy and fine-tune the target encoder on $(n_g, s_g)$ pairs from the downstream task's training set. To avoid over-fitting, we add an auxiliary loss which enforces the similarity between our target embedding and the embedding of the ground truth node containing the sentence needed to resolve the claim or question. Training and evaluation on FEVER and NQ required alignment between our version of Wikipedia and the one used to compile the benchmark and details of this can be found in Appendix C.1 and C.2.

Next, our navigation method requires a starting node. To put our agent close to the articles we need in the downstream tasks, we run a popular variant of TF-IDF, called BM25 [Robertson and Zaragoza, 2009]. We first create an index over all of the nodes in the graph, then take the top-5 BM25 matches as the starting nodes, and run navigation for 20 steps on each. Note that BM25 often does not find the exact text (i.e. the hits@1 is not high) but matches are in the right vicinity, usually the same or similar article. Thus the agent only has to navigate a few steps to reach the target: in FEVER and NQ datasets, the mean length of shortest path between start and target node is $4.3$ and $5.1$ steps respectively.

The final component is a re-ranker that takes all the sentences from all nodes visited by the agent and assigns a match score between each one and the claim. The ranked list of sentences can then be used for benchmark evaluation. We explore two types of ranker: (a) basic TF-IDF and (b) a transformer based model like BERT or BigBird, fine tuned using hard negatives mined from the TF-IDF ranker. See Appendix C.1 and C.2 for more details.

Evaluation of our approach thus consists of: (i) given a claim/question, use BM-25 to return a list of promising start locations; (ii) from the top $N = 10$ of these, run our navigation model for 20 steps, using the target encoder to embed the claim/question, (iii) use the re-ranker to score sentences visited against the claim/question and (iv) select the top 5 for computing the recall, precision, or F1 score, for comparison on the benchmarks.

## 5 Experiments

### 5.1 Investigation of training trajectory distribution

On the smaller 200k node graph, we investigate different choices for the trajectory policy distributions $\mathcal{E}$ described in Section 3: (i) **Random Forward Trajectories**, where we randomly sample start nodes and do a random walk to get a target, (ii) **Reverse Trajectories**, where we randomly sample a target node and and do a random walk on the reverse graph to get a start node, and (iii) **Random Shortest Paths** where we randomly sample a start node and take a random walk and then compute the shortest path between the start and target nodes. For each of these we trained policies on 5, 10, and 20 step navigation tasks. We evaluate these in two different ways: (i) forward sampling of source and target in the disjoint evaluation graph, (ii) reverse sampling of source and target. The results of this experiment are shown in Table 1. Unsurprisingly, we can see that forward and reverse trained policies both do better when evaluated in the way that matches training. However, forward trajectory policies

Table 1: Different trajectory distributions on small graph (200k nodes)

| Trajectory Policy | Navigation (Forward) | | | Navigation (Reverse) | | |
|---|---|---|---|---|---|---|
| | 5 | 10 | 20 | 5 | 10 | 20 |
| Forward | 85.3 | 76.4 | 67.5 | 31.1 | 32.6 | 19.9 |
| Reverse | 2.3 | 0.5 | 0.2 | 91.3 | 87.9 | 87.3 |
| Shortest path | 86.7 | 85.0 | 84.6 | 74.3 | 31.6 | 28.4 |

generalize much better to reverse navigation. When we analyze the traces of these experiments, we see that in the reverse sampling, many target nodes are in "dead-ends" where many target nodes do not have long random trajectories to sample. This results in easier navigation (as we can see from higher accuracy in reverse trajectory trained policies on reverse navigation) and more extreme overfitting, leading to poor performance of reverse trajectory on forward navigation.

Not surprisingly, we also see that training on a shortest path trajectories leads to better performance both in the forward and reverse sampling. This advantage is especially seen in the 20-step navigation task. Ideally, we would always train on shortest paths. However, the time complexity of pre-computing all shortest paths in a graph using Dijkstra's algorithm is $O(V(E + V log V))$ for the number of nodes $V$ and number of edges $E$ in the graph. On the full 38M Wikipedia graph, this computation would be completely intractable. Luckily, the random forward trajectories perform almost as well and can be computed in constant time with respect to the size of the graph and only linear time with respect to the trajectory length. For all subsequent experiments, we use the **Random Forward Trajectories** for our trajectory policy and evaluate on the forward version of the navigation tasks, where we call our approach **Random Forward Behavioral Cloning (RFBC)**.

## 5.2 Navigation

We next do a more thorough set of experiments on the smaller 200k Wikipedia graph. Because the graph is much smaller than the full Wikipedia and thus prone to overfitting, for all of our methods and baselines we use the fixed RoBERTA embeddings $\phi(s_i)$, the simplest 1-layer feed-forward network, and MiniBERT [Turc et al., 2019] for target sentence embedding $\phi(s_g)$.

Alongside our approach, we employ the following baselines/ablations:

- RFBC: Random-forward behavioral cloning (our approach).
- RFBC + RF: RFBC but with random features sampled from the unit sphere instead of $\phi$
- RL + RF: RL with random features.
- Random: policy that chooses random action (out-link) at each timestep.
- Greedy: selects the action embedding which has the smallest cosine distance with the target.
- Random DFS: DFS of depth equal to number of steps, choosing actions at random.
- Greedy DFS: DFS but select action with smallest cosine distance to the target.

More training details are in Appendix B.2 and computational cost of each in Appendix E.

The results evaluated on the held-out graph are shown in columns 1-4 of Table 2. Our approach (RFBC) performs well, achieving 77% in the multistep case. Performance also drops as the distance to the target increases. In contrast, the RL agent performs poorly (around 40% on navigation tasks). Because we generate virtually infinite expert trajectories with RFBC, RL performs worse as it is trained with essentially the same amount of exploration, but with a more sparse signal than behavioral cloning. Using random features in place of $\phi(.)$ caused the performance to drop to near chance, showing that the models are utilizing the semantics at each node and are not relying on a general search strategy of some kind. Random methods also do quite poorly, even when adding a DFS. Greedy methods do a bit better than random, especially with DFS, but still well below our method.

We also consider the more challenging sentence search task described in Section 4.1. This task requires not only finding a known target node, but learning to find a target node given only a single random sentence from that node, more closely matching our downstream task and the way humans tend to approach finding information on the web. Columns 5-7 of Table 2 show performance on this task for our approach and RL. Our method has reduced performance on the task due to the extra difficulty of learning the target embedding, but still performs reasonably. RL, however, degrades to near chance performance.

Table 2: Small graph navigation (200k nodes) – Success rate (%)

| Method | Navigation | | | | Sentence Search | | |
|---|---|---|---|---|---|---|---|
| | 5 | 10 | 20 | multistep | 5 | 10 | 20 |
| RFBC (ours) | 85.3 | 76.4 | 67.5 | 77.4 | 60.7 | 47.6 | 34.6 |
| RL | 41.4 | 40.2 | 41.6 | 43.6 | 14.0 | 1.5 | 8.9 |
| RFBC + RL (ours) | 85.1 | 77.6 | 68.1 | 78.3 | - | - | - |
| RFBC + RF | 17.4 | 16.3 | 16.6 | 17.1 | - | - | - |
| RL + RF | 6.0 | 7.5 | 7.0 | 6.7 | - | - | - |
| Random | 12.3 | 14.9 | 12.3 | 14.0 | - | - | - |
| Greedy | 19.7 | 16.7 | 21.7 | 23.4 | - | - | - |
| Random DFS | 10.0 | 9.5 | 8.3 | 10.0 | - | - | - |
| Greedy DFS | 31.1 | 23.8 | 22.7 | 51.8 | - | - | - |

Table 3: Wikipedia graph statistics

| Year | # Articles | # Nodes | # Edges | # Words / Node | Median Path Length |
|---|---|---|---|---|---|
| 2017 | 4.92M | 36.3M | 359M | 110 | 15 |
| 2018 | 5.27M | 38.5M | 387M | 109 | 15 |

## 5.3 Navigation on full 38M node Wikipedia graph:

Our method scales naturally to much larger graphs, which we demonstrate by training on the entire Wikipedia 2017 graph. We evaluate on the entire Wikipedia 2018 graph. Each graph is $\sim$ 37M nodes and $\sim$ 370M edges (see Table 3 for details). We would like to point out there is significant evolution (difference) between the 2017 and 2018 graph as analysed in Appendix A.3 along with further statistics.

We explore both architectures described in Section 4.2 for the policy network: (i) the single feed-forward layer (as used in the smaller graph) and (ii) the 4-layer Transformer model. We also compare using a fixed RoBERTA model for $\phi$ versus pre-training a text transformer directly on the navigation task (see Appendix B.3). DistillBERT [Sanh et al., 2019] is used for $\phi(n_g)$.

Table 4 shows the results on the full graph, evaluating on $\{5, 10, 20\}$ step tasks, for both navigation and sentence search. Figure 2 shows example trajectories of the trained agent. The first thing to note is that the overall navigation and sentence search performance of our methods is high. The 2018 Wikipedia graph we use for evaluation contains over 5 million articles, over 38 million nodes and 387 million edges. On this graph (which we generalize to from the previous year's graph) our best method can find an article 20 steps away 92.2% percent of the time and can locate it given just a single sentence of that article 90.2% of the time. This compares favorably with a similar evaluation performed in Nogueira and Cho [2016], where their WebNav approach achieved a 12.5% success rate for 16 step tasks on a 12M node graph. Compared to the performance of our method on the smaller 200k graph, we see that performance has greatly improved, most likely due to the addition of orders of magnitude more training data. In particular we see that sentence search performance on 20-step improves greatly from 34.6% to a best performance of 90.2%.

Additionally, in all cases, our learned embeddings perform better than using the fixed features, derived from large language models. In general, feed-forward policies perform best, except for 20 step navigation tasks. For longer trajectories, the Transformer policy network achieves slightly better performance, while on sentence search, the feed-forward model is consistently superior.

Table 4: Full graph navigation (38M nodes) - Success rate (%)

| Embedding + Policy | Navigation | | | Sentence Search | | |
|---|---|---|---|---|---|---|
| | 5 | 10 | 20 | 5 | 10 | 20 |
| RoBERTA + Feed-forward | 85.5 | 80.1 | 71.5 | 91.2 | 88.9 | 77.5 |
| RoBERTA + Transformer | 85.3 | 88.3 | 87.9 | 81.6 | 75.4 | 70.9 |
| Embed train (ours) + Feed-forward | 96.1 | 94.1 | 89.8 | 96.3 | 92.8 | 90.2 |
| Embed train (ours) + Transformer | 93.5 | 90.6 | 92.2 | 93.9 | 86.3 | 79.7 |

## 5.4 Application to fact verification

We now evaluate the ability of our navigation approach to select evidence on the FEVER development set [Thorne et al., 2018] with results shown in Table 5. The primary metric used is F1@top5 but we also give precision and recall. Paired with a basic retrieval (BM25) and re-ranker (TF-IDF), the approach obtains a respectable F1 of 0.46. If we remove our navigation component then this drops substantially to 0.36, demonstrating its utility. Swapping to a more powerful re-ranker (BigBird) boosts the F1 to 0.731, which is competitive with the state-of-the-art (FEVER leaderboard [2022] rank #6; rank #1 F1 is 0.799). We also compare to a leading approach [Stammbach, 2021] that similarly relies on the BigBird re-ranker, with our approach having a significantly better F1 score.

Table 5: Results on the FEVER benchmark (evidence retrieval only). The first two rows show the effect of adding our navigation scheme to the simple BM25 retrieval and TF-IDF re-ranker combination. Switching to a more powerful re-ranker (BigBird) results in a significant boost in F1 score, surpassing Stammbach [2021] who also use BigBird [Zaheer et al., 2020].

| Method | Precision@5 | Recall@5 | F1@5 |
| --- | --- | --- | --- |
| BM25 + TF-IDF [Thorne et al., 2018] | 0.33 | 0.40 | 0.36 |
| BM25 + RFBC (Ours) + TF-IDF | 0.38 | 0.55 | 0.46 |
| BM25 + RFBC (Ours) + BigBird | 0.71 | 0.75 | 0.73 |
| Stammbach [2021] | 0.26 | 0.94 | 0.41 |

Figure 2: Example navigation trajectories in the 38M node Wikipedia graph. The start and target nodes are shown in the first two rows. Parentheses indicate paragraph block (zero-indexed) within article. Note that in cases where the agent fails to find the target node (cols 1 & 3), it visits ones that are very close by.

Figure 3: Successful navigation trajectories on the FEVER [Thorne et al., 2018] evaluation set. Rows from top: claim to be verified; starting node (from BM25); ground truth target (not visible to agent); target node text; agent trajectory.

A possible explanation is that their approach overwhelms the re-ranker with all possible hyperlinks whereas, navigation being inherently sparse, our model presents a more refined set for re-ranking. We note that our approach: (a) is significantly simpler than many top ranked approaches and (b) selects evidence over a much larger set than the curated version of Wikipedia used in FEVER (38M vs 5M). Examples of successful navigation traces are shown in Figure 3.

### 5.5 Application to question answering

Finally, we present results on another downstream task where navigation helps. In particular, we consider the task of finding correct evidence passage for open domain question answering. We use the Natural Question (NQ) open-domain dataset presented in Lee et al. [2019]. Since we target navigating to the exact evidence passage required to answer the question, we use recall@{1,2,3,4,5} for finding the gold evidence passage as our metric. More commonly the metric marks a retrieved passage to be correct if it contains the answer string, but this causes a lot of false positives, e.g. the answer string appears in a totally irrelevant context.

Table 6: Results on the Natural Questions open domain benchmark (evidence retrieval only). Third and fourth rows show the effect of adding our navigation scheme to the simple BM25 retrieval and BigBird re-ranker. First row is a reference re-training of state-of-the-art method on our setup.

|  | Recall@1 | Recall@2 | Recall@3 | Recall@4 | Recall@5 |
|---|---|---|---|---|---|
| RocketQA [Qu et al., 2021] | 32.7 | 43.8 | 51.7 | 58.3 | 62.5 |
| BM25 | 11.1 | 17.3 | 21.8 | 24.9 | 27.3 |
| BM25 + BigBird ReRank | 22.2 | 30.2 | 35.1 | 38.7 | 41.2 |
| BM25 + RFBC (Ours) + BigBird ReRank | 31.6 | 41.3 | 46.6 | 49.8 | 51.6 |
| Improvement | +42.3% | +36.7% | +32.7% | +28.7% | +25.2% |

The results are tabulated in Table 6. Paired with a basic retrieval (BM25) and re-ranker (BigBird), the approach obtains a respectable Recall@1 of 31.6. If we remove our navigation component then this drops substantially to 22.2, demonstrating its utility. As a reference, we also ran a state-of-the-art system RocketQA [Qu et al., 2021] on our setup and evaluated it using our harder metric. It is worthwhile to note that our approach (a) is significantly simpler than state-of-the-art approaches like RocketQA which employ 4 stages of dual-encoder and cross-attention models, and (b) selects evidence over a larger set than the commonly used preprocess Wikipedia passages (38M vs 21M).

## 6 Discussion

We have presented a simple and effective scheme for navigating a large Wikipedia graph that is applicable to more general web navigation problems. We show that behavioral cloning of random trajectories is a viable approach to learning both entity embeddings and a navigation policy. When applied to the fact verification task on FEVER dataset and the open-domain question answering task on NQ dataset, they offer highly competitive performance, whilst being complementary to existing approaches. Another advantage worth highlighting is that the navigating agent provides a provenance on how it arrived at the relevant evidence, which many other methods (like dense passage retrieval) do not provide. One limitation of our approach when we move from Wikipedia to the wider Internet is that our scheme relies on a good target encoder. For navigation and our downstream tasks, there was a clear ground-truth target node available training, but in other settings this might not be the case. Another limitation is that we require a re-ranker to decide on the final retrieved sentence, but ideally the agent would decide for itself when it has reached the correct node.

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
