# A Data / Environment details

## A.1 Wikipedia data processing

We start by downloading English Wikipedia snapshot (the pages-articles.xml.bz2 file) from Wikimedia[3] or Internet Archive.[4] We extract text from English Wikipedia for a given snapshot using Gensim's WikiCorpus class.[5] For each page, this tool extracts the plain text and hyperlinks, and strips out all structured data such as lists and figures. To avoid pages without sufficient textual content, we filter out categories, listical, disambiguation pages and any other page with less than 200 characters. Each article ($\mu = 1000$ words) is split into paragraph-sized blocks ($\mu = 100$ words), which form the *nodes* in our Wikipedia Graph. This was done both for conceptual reasons (humans don't read an entire article at once but look at bits and pieces of it at one go) and for modeling reasons (most language models have an upper input token length that is smaller than the average Wikipedia page). The *edges* in the graph are formed in three ways:

1. for nodes in the same article, we add "previous node" and "next node" links to connect them in a chain;

2. organic hyperlinks – the internal links connecting Wikipedia pages to each other; and

3. we additionally run entity linking[6] because repeat mentions of an entity in Wikipedia lack hyperlinks, which is problematic when articles are split up.

This constitutes our Wikipedia graph construction, where the text blocks act as vertices and the organic hyperlinks along with entity links are the edges. We store the graph as an adjacency list with metadata, using a memory mapped key-value data-structure to enable fast random access during navigation.

We run the above graph generation pipeline for two different snapshots of Wikipedia:

1. June 01 2017[7]: This corresponds to FEVER fact verification dataset.

2. December 2018[8]: This corresponds to Natural Questions open dataset.

## A.2 Smaller 200k node graph construction

We construct smaller train and eval graphs with 200k nodes each for the following reasons:

1. Checking strict generalization by making train and eval graphs to be totally disjoint, i.e. no common nodes or edges. The full 2017 and 2018 are sufficiently different but not entirely disjoint.

2. Running experiments on full graphs are expensive, so for more thorough evaluation across multiple baselines we constructed the smaller graphs.

To construct these two smaller graphs, we start with the full 2018 graph. We sort the nodes by their in-degrees. We mark nodes with odd ranks to be in train and even ranks to be in eval, which ensures complete separation between the two. For the train graph we start with rank 1 node (and for eval with rank 2 node) and then select all its neighbors which have odd rank (even rank). Then all the odd (even) nodes connected to the selected ones are picked. This process is continued until the desired number of nodes are selected.

---

[3]https://dumps.wikimedia.org/enwiki/

[4]https://archive.org/search.php?query=Wikimedia%20database%20dump%20of%20the%20English%20Wikipedia

[5]https://github.com/RaRe-Technologies/gensim/blob/master/gensim/corpora/wikicorpus.py

[6]We use a standard linker from https://cloud.google.com/natural-language/docs/analyzing-entities

[7]https://archive.org/download/enwiki-20170601/enwiki-20170601-pages-articles.xml.bz2

[8]https://archive.org/download/enwiki-20181220/enwiki-20181220-pages-articles.xml.bz2

## A.3 Graph statistics

We construct two versions of the full Wikipedia graph following the method outlined in Appendix A.1. The main summary statistics (number of articles, nodes, edges, words per node, median path length) of the resulting graphs are listed in Table 3.

A more detailed picture of the graph can be obtained by looking at its degree distribution, which is presented in Figure 4. It shows the graph has tell-tale sign of web-graph: it has a power-law distribution with a few hub nodes.

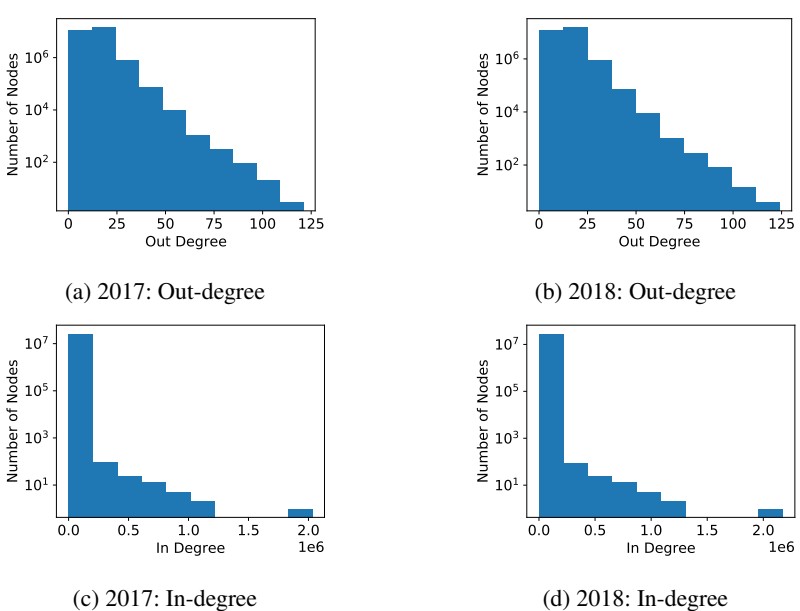

(a) 2017: Out-degree        (b) 2018: Out-degree

(c) 2017: In-degree        (d) 2018: In-degree

Figure 4: Degree Distribution

We try to estimate how far are two nodes in the graph. Calculating the entire distribution of shortest path length is prohibitively expensive for such large graph. We follow Ye et al. [2010] method to get an approximation. We start by selecting 250k nodes having highest in-degree, which is a good approximation of the giant component in a web graph as required by Ye et al. [2010]. We then perform a single source to all shortest paths on each of these 250k nodes using Dijkstra's algorithm for sparse graphs. This resulted in computing approximately 1 trillion shortest paths based on which the estimated distribution of shortest path length is shown in Figure 5.

Finally, we would like to point out that there is significant evolution (difference) between the 2017 and 2018 graph. Some important statistics reflecting the changes across the two graphs are:

- New articles: 404,773 (7.5%)
- New nodes: 3,871,578 (10%)
- Deleted articles: 58,476 (1.1%)
- Deleted nodes: 874,372 (2.3%)

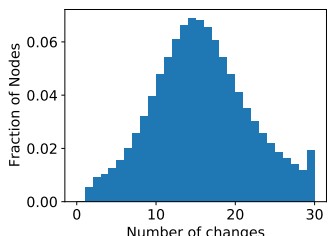

Figure 6: Number changes in edges among common nodes between 2017 and 2018

Further in Figure 6, we show histogram of changes in edges among the common nodes between the two graphs. Apart from addition/deletion of organic hyperlinks, just modifications of text would yield to differences as the chunking will be different. This shows significant generalization is needed to successfully navigate across the two graphs, simply memorization will not work.

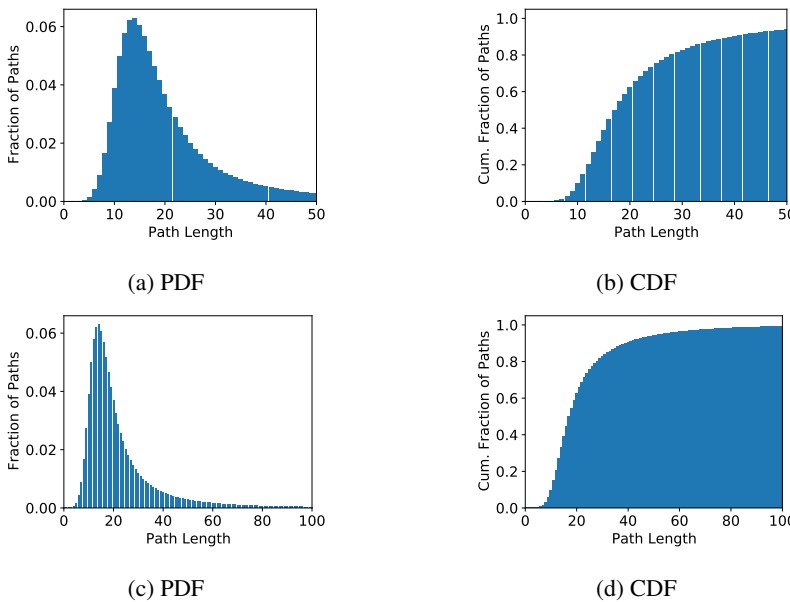

|  |  |
|:-:|:-:|
| (a) PDF | (b) CDF |
| (c) PDF | (d) CDF |

Figure 5: Distribution of Short Path Length estimated by computing shortest path between 1 trillion pairs of random nodes in the 2018 graph.

# B    Experimental details

## B.1    Embedding pretraining

This section details our embedding pretraining procedure, as introduced in Section 3.

**Model details**    The transformer is fine-tuned from a pre-trained RoBERTa [Liu et al., 2019] model. As in Equation 1 our model is trained to optimize

$$\mathcal{L}(\theta) = \mathbf{E}_{\mathcal{E}(n_0,...,n_T)} \left[ \sum_{t=1}^{T-1} \log p_\theta(n_t|n_t, n_g) \right] \tag{5}$$

where we set $n_g = n_T$. We parameterize this distribution as

$$p_\theta(n_{t+1}|n_t, n_g) \propto \exp(f_\theta(s_\phi(n_g), s_\phi(n_t), a_\phi(n_t, n_{t+1}))) \tag{6}$$

where $s_\phi(\cdot)$ and $a_\phi(\cdot, \cdot)$ are functions which extract embeddings for an entire node and a node-action combination, respectively and $f_\theta(\cdot, \cdot, \cdot)$ is a function which combines these embeddings to produce action probabilities. At their core, $s_\phi$ and $a_\phi$ are based on the same transformer model. We tokenize the text at node $n$ to produce $L$ tokens and pass these tokens through the transformer to produce $L$ embeddings.

The state representation $s_\phi(n)$ is produced by taking the mean of these embeddings and passing this vector through a `tanh` nonlinearity. The node-action embedding $a_\phi(n, n')$ embeds the action of moving from node $n$ to its neighbor $n'$. Let us assume the text at node $n$ is "Barack Obama was the President of the United States during..." where the substrings "Barack Obama" and "President of the United States" correspond to hyperlinks to other neighboring articles $n'$ and $n''$. Then we construct the node-action embedding $a_\phi(n, n')$ by passing the tokenized text of node $n$ through our transformer, again producing $L$ transformed token embeddings. Then we take the transformed tokens which correspond to the text "Barack Obama" and take their mean and pass this through the `tanh` nonlinearity. If we instead wanted to produce $a_\phi(n, n'')$ we would take the transformed tokens which correspond to the text "President of the United States" instead, take the mean and apply the nonlinearity. This is visualized in Figure 7.

The function $f_\theta$ which combines these embeddings first concatenates them and passes the combined input through a simple neural network with a single hidden layer and 1 output neuron.

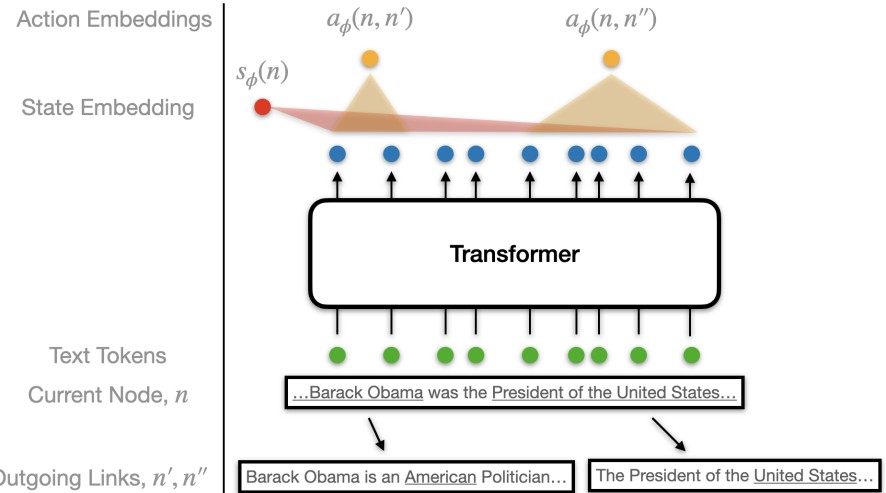

Figure 7: Description of how embeddings are extracted for navigation pretraining.

**Training details**   We train these models to optimize Equation 5. The objective is optimized with the Adam [Kingma and Ba, 2014] optimizer using an initial learning rate of $10^{-5}$ with a one-cycle cosine decay schedule and linear warm up of 10K steps. The training batch size is 512 and the model is trained over 3M iterations. Node text is clipped (or padded) to 200 tokens. The weights of the transformer are initialized from a pretrained RoBERTa [Liu et al., 2019] masked language model.

## B.2   Small graph experiments

**Model details**   In the small graph experiments, we use the simple policy network $FF$ as described in Section 4.2. As discussed, we use a 1-layer MLP (meaning a single learned linear layer with no activation function). We use a hidden size of 768 to match the size of $\phi$. Because the final logits are the result of an inner product between each state and action embedding of size 768, to aid network training, we add a normalization layer (jax.nn.normalize) before the inner product.

For the action embedding $a_i = \phi(s_i)$ we also concatenate a 1-hot vector that represents the action type (e.g. whether it was a link action, or a next action, or a prev action). We also concatenate another bit that indicates whether the state $s_i$ of the particular action has been visited before.

For the purposes of the RL baseline, we can view graph navigation as a goal-conditioned MDP.[9] Our observation is the same as the inputs to the RFBC models. The reward function $R$ for current node $n$ and goal node $n_g$ we can write simply as:

$$R = \begin{cases} 1 & n = n_g \\ 0 & otherwise \end{cases} \qquad (7)$$

For the RL baseline, we used IMPALA [Espeholt et al., 2018], which is a kind of policy gradient approach, and thus has a policy network $\pi$ and a "baseline" or value function network. Since the policy network has the same output space as our RFBC models, we use an identical architecture for this. For the value network, we use a very similar architecture. We again use $\phi$ as our encoding of the current node and the target node, again concatenate them together. We then feed this through a 1-layer MLP to compute the value.

**Training details**   The procedure for generating the trajectories we use for the BC loss is described in Section 3.1. Except for our experiment in Section 5.1, we always use the random forward trajectories. Following Lin et al. [2018], in the navigation experiments, we randomly drop out edges in the training graph with probability 0.5 to reduce over-fitting. As stated in Section 4.1, we set max steps

---

[9]On a known fixed graph, we can formulate this an an MDP. However, in the case where the graph changes or in a setting where there are multiple possible graphs it is trivial to reformulate this as a POMDP where we only can see the current node and neighboring nodes

to $B = 100$. For RFBC training on the small graph, we use RMSProp with a learning rate of $0.01$, a decay of $0.9$ an epsilon of $10^{-10}$.

For the sentence search tasks, our models train a separate $\phi$ for the target embedding. As we stated, we use MiniBERT for this embedding $\phi_{\text{target}}$ and train it with the same optimization settings, except that we reduce the learning rate for these weights to $10^{-4}$. We use a batch size of $512$ and we train for $50,000$ network update steps.

For RL training, we use VTRACE [Espeholt et al., 2018] loss, using the reward above. We again use RMSProp with a learning rate of $0.01$, a decay of $0.9$ an epsilon of $10^{-10}$. We set the baseline cost to $0.5$, trajectory length (the number of timesteps the RL loss backprops through) to $100$, batch size again to $512$ and max update steps to $50,000$. We did a parameter sweep for entropy costs of $0.1, 0.01, 0.001$ and gamma of $0.8, 0.9$. To give RL the best chance possible, we choose the maximum over this sweep for each experiment (but still none of these match RFBC performance).

### B.3 Full Wikipedia navigation experiments

On the full Wikipedia training experiments in Section 5.3, we evaluate on navigation and sentence search using either our feed-forward or transformer model and using either RoBERTA fixed embeddings, or our trained embeddings described in Appendix B.1.

The feed-forward model is identical to the one described in Appendix B.2. The transformer model is the standard transformer model from Vaswani et al. [2017] with 4 layers, an attention size of 64, 12 heads, and mlp hidden size of 3072 and dropout rate of 0. DistillBERT [Sanh et al., 2019] is used for $\phi(n_g)$ for the sentence search experiments.

The learning parameters and all other relevant training parameters for all models are identical to those in Appendix B.2 except that the learning rate for the transformer model is lower ($10^{-4}$) as the earlier learning rate was too high for transformer models.

### B.4 RFBC + RL

We additionally try finetuning an RL policy starting from our RFBC-trained navigation policies. We use the same network and learning settings as we did in the other RL experiments. We train for an additional 10,000 network updates (512M environment steps). We add the final numbers in Table 2 and show the training curve in Figure 8. We can see from the training curve that RL training accuracy fluctuates around the point where RFBC training left off. From the results in Table 2 we see little statistically significant difference (about 1 point in either direction).

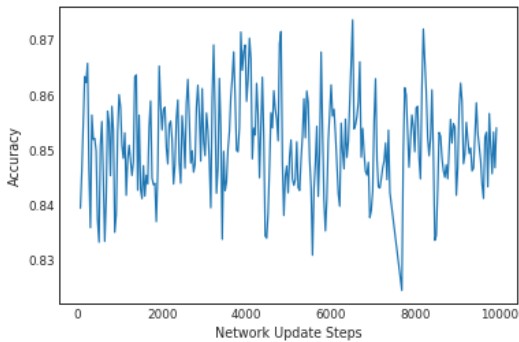

Figure 8: Training curve for RFBC with RL finetuning.

## C  Downstream task details

### C.1  FEVER experiments

FEVER [Thorne et al., 2018] is a dataset containing $185,445$ claims labeled as SUPPORTED, RE-FUTED, or NOTENOUGHINFO. Verifiable claims (i.e. SUPPORTED or REFUTED) are annotated with

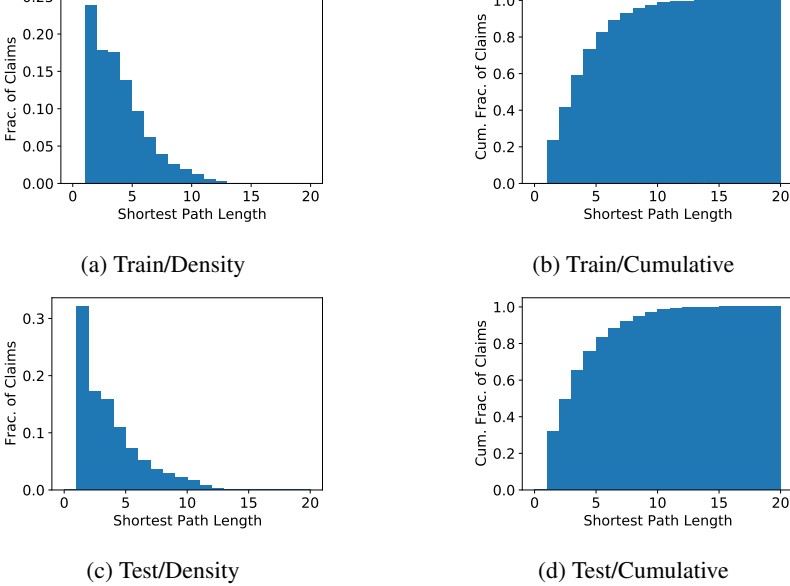

(a) Train/Density

(b) Train/Cumulative

(c) Test/Density

(d) Test/Cumulative

Figure 9: Distribution of Short Path Length from starting node computed by BM25 to target node on FEVER dataset on 2017 graph.

evidence sentences supporting this classification, which are drawn from a preprocessed version of the June 2017 Wikipedia snapshot. Notably, only the article introductions are retained, and claims are generated from a curated set of approximately $50,000$ popular pages.

**Data processing**    To make an aligned navigation graph for FEVER, we start with the June 01 2017 snapshot of Wikipedia and build the graph using the procedure described in Section A.1. Because of differences in text preprocessing, FEVER's version of Wikipedia does not precisely align to our own. We then try to match FEVER's Wikipedia articles to our graph, relying on Wikipedia URL redirections for handling disambiguations. This resulted in $99.5\%$ matching of articles between our navigation graph and the FEVER version of Wikipedia. Second, we align the evidence sentences in FEVER to sentences from our text blocks using a fuzzy string match. Specifically, we use the token set ratio to score the similarity between an evidence sentence and the text in a graph node. If the score between a sentence and a node is $\geq 80$, the node is added as an evidence node for the given claim. This threshold is sufficiently high to minimize the chances of a false positive, while also matching a high percentage of evidence sentences. Some sentences are difficult to match, particularly those that are split between two text blocks. Ultimately, $93.1\%$ of all evidence sentences were matched to a node. This gives us an augmented FEVER dataset, where for each claim we have a corresponding set of evidence nodes in the graph. These nodes are then used as navigation targets for fine-tuning during the training phase. We generate trajectories for BC by running BM25 over all nodes in the graph, taking the top 10 matches as starting nodes, and finding shortest paths to evidence nodes.

**Data statistics**    We compute the shortest-path distance between top-1 retrieval of BM25 and target node in this graph in Figure 9. As can be seen that most path lengths are relatively short, which implies that BM25 lands us in the right vicinity and by a small amount of navigation around we will be able to find the right evidence passage.

**Training details**    For the FEVER benchmark, our model uses learned embeddings for $\phi$ with a single feed-forward layer for the policy network. The model is pretrained on the 5-step sentence search task, and finetuned on trajectories generated from the augmented FEVER dataset using the the following loss function:

$$\mathcal{L}(\theta) = \mathcal{L}_{\text{BC}}(\theta) + 0.1\|\phi_{\text{target}}(\text{claim}) - \phi(s_g)\|_2 \tag{8}$$

where $\mathcal{L}_{\text{BC}}$ is the normal BC loss we use in our other experiments.

Only the $\phi_{\text{target}}$ weights are finetuned for this task. We use AdamW with a constant learning rate of $10^{-6}$, $\beta_1 = 0.9$, $\beta_2 = 0.999$, and $\epsilon = 10^{-6}$. The model is finetuned for $100,000$ update steps with a batch size of $512$. No edge dropout is used during finetuning.

**Evaluation details**  For each claim in the FEVER development dataset, we first run BM25 over all nodes in the graph and take the top 5 matches as starting nodes. From each node, the finetuned model then navigates for 20 steps. We collect all sentences in all nodes visited by the agent over the 100 total navigation steps, and match them to FEVER evidence sentences using the WRatio function in the fuzzywuzzy package.[10] Two sentence ranking methods are explored: Gensim's TF-IDF implementation,[11] and the BigBird re-ranker of Stammbach [2021], using their open source model and weights.[12] Finally, the top-5 evidence sentences are then submitted to the official FEVER scorer[13] to compute the accuracy, recall, and F1 scores.

## C.2  NQ experiments

Natural Questions (NQ) is a dataset collected from real users asking questions on the Google search engine which are answerable using Wikipedia. We use the Natural Question open-domain subset presented in Lee et al. [2019] which has been aligned to Wikipedia dump of December 20, 2018 by Karpukhin et al. [2020]. In this split it has 58,880 questions for training and another 8,757 questions as development set for evaluation. (The test set is not aligned to Wikipedia passages so we do not evaluate on it.)

**Data processing**  Most prior works in literature utilize the processed collection of 21M passages provided by Karpukhin et al. [2020], but unfortunately we cannot use it as it has no hyperlink information which is crucial for our graph building. So we re-align the questions in NQ to our 2018 graph with 38M passages (i.e. nodes) which has different text blocks than Karpukhin et al. [2020]. As the article title corresponding to the passage was given as part of the dataset, the alignment task was local to the document. We could find all the article titles, i.e. 100% match in locating the document in our dump. As before, to align the evidence passage in NQ to our text blocks, we first used a fuzzy string match. Then in the ranked list of fuzzy string matches, we searched for exact answer string. The highest ranked node with answer string was labelled as the ground truth. Some answer strings were difficult to match, particularly those that are split between two text blocks. Ultimately, 99.3% of all evidence passages were matched to a node. This gives the re-aligned NQ dataset, where for each question we have a corresponding set of evidence nodes in our graph. These nodes are then used as navigation targets for fine-tuning during the training phase. We generate trajectories for BC by running BM25 over all nodes in the graph, taking the top 10 matches as starting nodes, and finding shortest paths to evidence nodes.

**Data statistics**  We compute the shortest-path distance between top-1 retrieval of BM25 and target node in this graph in Figure 10. As can be seen that most path lengths are relatively short, which implies that BM25 lands us in the right vicinity and by a small amount of navigation around we will be able to find the right evidence passage.

**Training details**  For the NQ benchmark, our model uses learned embeddings for $\phi$ with a single feed-forward layer for the policy network. The model is pretrained on the 5-step sentence search task, and finetuned on trajectories generated from the NQ dataset using the the following loss function:

$$\mathcal{L}(\theta) = \mathcal{L}_{\text{BC}}(\theta) + 0.1\|\phi_{\text{target}}(\text{question}) - \phi(s_g)\|_2 \tag{9}$$

where $\mathcal{L}_{\text{BC}}$ is the normal BC loss we use in our other experiments.

Only the $\phi_{\text{target}}$ weights are finetuned for this task. We use AdamW with a constant learning rate of $10^{-6}$, $\beta_1 = 0.9$, $\beta_2 = 0.999$, and $\epsilon = 10^{-6}$. The model is finetuned for $100,000$ update steps with a batch size of $512$. No edge dropout is used during finetuning.

---

[10]https://pypi.org/project/fuzzywuzzy/
[11]https://radimrehurek.com/gensim/models/tfidfmodel.html
[12]https://github.com/dominiksinsaarland/document-level-FEVER
[13]https://github.com/sheffieldnlp/fever-scorer/blob/9d9ed27637adddf73bc2f8e38c436bdc032c9f1f/src/fever/scorer.py

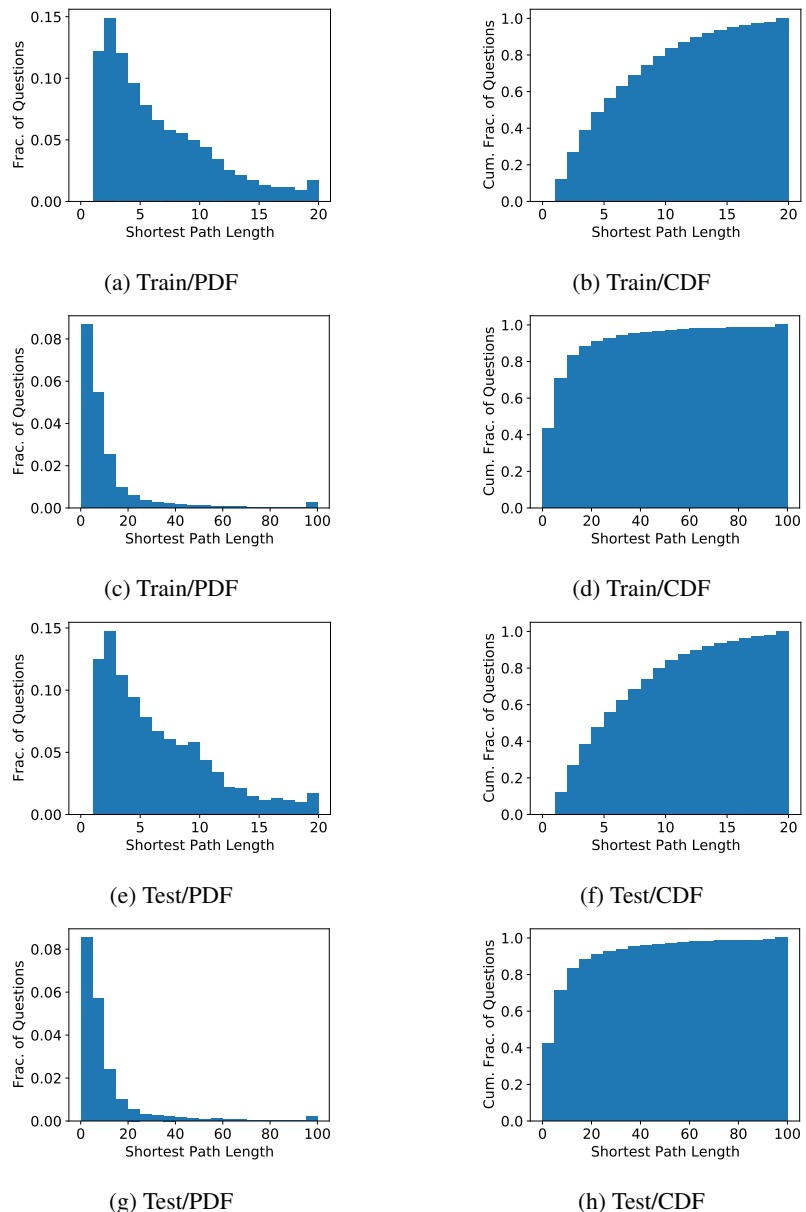

Figure 10: Distribution of Short Path Length from starting node computed by BM25 to target node on Natural Questions dataset on 2018 graph.

**Evaluation details** For each question, we first run BM25 over all nodes in the graph and take the top-5 matches as starting nodes. From each starting node, the agent then navigates for 20 steps. All the 100 nodes visited by the agent is then ranked by a simple cross-attention model. For the cross-attention model, we follow Hofstätter et al. [2020] to train 6-layer BigBird [Zaheer et al., 2020]. Since we target navigating to the exact evidence passage required to answer the question, we use recall@{1,2,3,4,5} for finding the gold evidence passage as our metric. More commonly the metric marks a retrieved passage to be correct if it contains the answer string, but this causes a lot of false positives, e.g. the answer string appears in a totally irrelevant context.

# D  Variational Interpretation of our Approach

We present an alternative motivation of our method based on variational inference in a latent variable model of states. Assume we are given some goal node $n_g$. We would like to parameterize a model $p_\theta(n_T|n_g)$ which will generate a trajectory of states $n_0, n_1, \ldots, n_T$ such that $n_T = n_g$. We define our model as a latent variable model

$$p_\theta(n_T|n_g) = \sum_{n_0, n_1, \ldots, n_{T-1}} p_\theta(n_T, n_{T-1}, \ldots, n_1, n_0|n_g)$$

$$= \sum_{n_0, n_1, \ldots, n_{T-1}} p(n_0) \prod_{t=1}^{T} p_\theta(n_t|n_{t-1}, n_g) \tag{10}$$

i.e an autoregressive model which samples an initial state from $n_0 \sim p(n_0)$ and then samples subsequent states $n_t \sim p_\theta(n_t|n_{t-1}, n_g)$. The transition distribution samples the next node $n_t$ from the neighbors of $n_{t-1}$ in our graph. We would like the probability that $n_T = n_g$ to be large, thus we will train our model to maximize $\log p_\theta(n_T = n_g|n_g)$.

For latent-variable models such as this, we can rewrite the marginal likelihood as

$$\log p_\theta(n_T|n_g) = \mathbf{E}_{p_\theta(n_0, \ldots, n_{T-1}|n_T, n_g)} \left[ p(n_0) \prod_{t=1}^{T} p_\theta(n_t|n_{t-1}, n_g) - \log p_\theta(n_0, \ldots, n_{T-1}|n_T, n_g) \right] \tag{11}$$

and can obtain a lower-bound on this quantity by replacing the intractable posterior $p_\theta(n_0, \ldots, n_{T-1}|n_T, n_g)$ with a variational approximation $q(n_0, \ldots, n_{T-1}|n_T, n_g)$, i.e

$$\mathcal{L}_\theta(n_T, n_g; q) := \log p_\theta(n_T|n_g)$$

$$\geq \mathbf{E}_{q(n_0, \ldots, n_{T-1}|n_T, n_g)} \left[ p(n_0) \prod_{t=1}^{T} p_\theta(n_t|n_{t-1}, n_g) - \log q(n_0, \ldots, n_{T-1}|n_T, n_g) \right]. \tag{12}$$

The above bound becomes tight when $q(n_0, \ldots, n_{T-1}|n_T, n_g) = p_\theta(n_0, \ldots, n_{T-1}|n_T, n_g)$. Thus, we can optimize our model parameters $\theta$ to maximize $\mathcal{L}_\theta(n_T, n_g; q)$. This approach has had a long history of successfully training latent-variable generative models [Kingma and Welling, 2013, Ho et al., 2020].

In the context of this work, we can view $q(n_0, \ldots, n_{T-1}|n_T, n_g)$ as a distribution over trajectories of nodes which end at our goal node. We can interpret the various trajectory generation methods introduced in Section 3.1 as different variational approximations $q$. Ideally, the option which most closely approximates the true posterior $p_\theta(n_0, \ldots, n_{T-1}|n_T, n_g)$ would be the most desirable but in general this distribution is intractable.

We find that using simple random trajectories provides a good-enough approximation to the posterior to enable us to train a model which reliably finds the goal state. This result is not completely surprising in context of prior work [Dai et al., 2020] which successfully trains latent-variable models using random trajectories as an inference model.

# E  Efficiency Analysis

In this section we analyze the asmpytotic runtime of our method and baselines. In Table 7 we show the asymptotic runtime and whether or not it is scalable for each method. We also show the 5-step accuracy from Table 2 for quick reference.

$T$ is the trajectory length (either the length of the trajectory for BC methods or the maximum allowed trajectory for other methods. For BC, $T$ is guaranteed to be shorter than the max, but this is a constant factor difference. $E_{out}$ is the average number of outgoing edges (i.e. actions) at a node. (small - $10^1$) $E$ is the total edges in the graph (very large - $10^8$). $V$ is the number of nodes in the graph (large - $10^7$).

Table 7: Asmptotic runtime analysis of RFBC and baselines.

| Method | Cost (Big-O) | Scalable | Accuracy (5-step 200k graph) |
|---|---|---|---|
| RFBC (ours) | $O(E_{out}T)$ | Yes | 85.3 |
| Backwards BC | $O(E_{out}T)$ | Yes | 2.3 |
| Shortest path BC | $O(E \log V + E_{out}T)$ | No | 86.7 |
| RL | $O(E_{out}T)$ | Yes | 41.4 |
| Random | $O(T)$ | Yes | 12.3 |
| Greedy | $O(E_{out}T)$ | Yes | 19.7 |
| Random DFS | $O(T)$ | Yes | 10.0 |
| Greedy DFS | $O(E_{out}T)$ | Yes | 31.1 |

| | |
|---|---|
| $V$ | nodes in the graph [large $- 10^7$] |
| $E$ | total edges in the graph [very large $- 10^8 - 10^9$] |
| $E_{out}$ | average out degree of node (# of actions) [small $- 10^1$] |
| $T$ | length of trajectory, i.e. the maximum steps agents can run. [small $- 10^1$] |

For most methods, the runtime is $O(E_{out}T)$ because the method does some constant amount of work for each possible actions $E_out$ for $O(T)$ steps. For the random methods it is $O(T)$ because the work is constant per step (choose randomly). Shortest path distance has the worst runtime. The $O(ElogV)$ term is the runtime of finding a shortest path (Dijkstra's algorithm with Fibonacci heap). As the graph gets larger, this term because infeasible (we could not even run this on the full Wikipedia graph). DFS might seem like it should be larger than $O(E_{out}T)$ or $O(T)$, but because we restrict the maximum allowable steps, it is only proportional to $T$ and (for greedy) $E_{out}$.

From this Table, we see that all methods (except shortest path) have scalable runtimes. The dominant factor is $E_{out}$, which theoretically grows slowly with graph size. Moreover, in practice, there is a natural upper bound on the number of links one can have in a paragraph of text, and the runtime difference in our experiments going from 200k nodes to 38M nodes is only 1.5x despite the 200-fold increase in graph size.