# OpenReview forum: "Learning to Navigate Wikipedia by Taking Random Walks"
_NeurIPS.cc/2022/Conference — NeurIPS 2022 Accept_

### Official Review · Reviewer_L8iM · 2022-07-09

**Rating:** 5
**Confidence:** 3
**Soundness:** 3 good
**Presentation:** 3 good
**Contribution:** 2 fair

**Summary:**

The paper presents a web navigation agent that learns to reach target webpages by hopping hyperlinks. The paper uses Transformer-based models that encode the local content and learn to select the link the agent moves to next. This model is trained by behavior cloning of randomly sampled trajectories. The authors show that this approach provides improved performance on navigation and fact verification benchmarks.

**Questions:**

- I wonder if the authors have insights into why RL performs more poorly than the proposed RFBC (Table 2)?
- Suggestion for related work discussion: there is a larger body of existing works in machine learning for Wikipedia hyperlink graphs, which may be worth discussing -- e.g. https://arxiv.org/abs/2203.15827 (Transformer language model that predicts hyperlinked pages from the current page), https://arxiv.org/abs/1911.10470 (QA model that predicts hyperlinked pages). The main difference of the proposed work from these papers seems that it focuses on longer-horizon navigation with a target in consideration.


**Limitations:**

- The authors discusses the limitation in the paper.

**Strengths And Weaknesses:**

Strengths
- The proposed system is clean, and performs better than existing systems on navigation and fact verification benchmarks
- Good execution - the paper conducts various ablation studies such as different navigation setups and different design choices of the method (trajectory sampling, policy features, etc.)

Weaknesses
- This might be due to the presentation of the paper, but the technical novelty or contribution appears somewhat limited to me after reading the paper. The method seems to be a fairly simple application of existing techniques such as Transformer encoder and behavioral cloning training. The experiments also seem not quite offering significant insights or takeaways for the Neurips community. It would be really helpful if the authors could crisply clarify the main contribution with respect to existing works, e.g. is it a new methodology, or a new empirical finding (if so, what might be the impact to the research community), or an improved performance (if so, what exact difference from prior works is enabling the improved performance)?
- The navigation experiments are only conducted on a Wikipedia graph, which is smaller than the real-world web

---

> ### Author Response · Authors · 2022-08-02
> **Response**
>
> We thank the reviewer for the detailed comments and for recognizing that our proposed navigation method is clean, effective, and well-executed. We also would like to point the reviewer’s attention towards an additional task of information gathering for question answering on Natural Question (NQ) benchmark provided in the Appendix C.2 on which we obtain strong performance.
>
>
> > Technical novelty or contribution
>
> We apologize for the not making the main contributions of the work clear and have revised the paper accordingly:
> * Our work addresses the problem of web navigation, central to the building of intelligent web agents/assistants. Hitherto, web agents received minimal attention from the ML community but now is a hot topic, with papers such as WebGPT, GopherCite, LaMDA-research, REINA, etc.
> * We introduce the first viable solution to this problem of long-horizon web navigation, showing generally reliable (~90% success) navigation on a graph of size 10^7. Previous attempts such as Nogueira & Cho (2016) show a far inferior performance, well below that needed for practical utility.
> * Since there is nothing Wikipedia-specific about our approach and it scales gracefully (see response to reviewer J6Fp), the method is applicable to more general web settings.
> * Our paper also demonstrates the relevance of the pre-train + fine-tune paradigm to the web domain, showing that pre-training for navigation is a suitable objective to aid downstream tasks with limited data such as Q&A.
> * This is demonstrated by applying our navigation approach on two challenging Q&A tasks (FEVER and Natural Questions). In both settings, we show a jump in performance over existing search-based approaches that rely on retrieval + reranking, matching SOTA methods despite using far simpler retrieval and reranking mechanisms.
>
>
> > The navigation experiments are only conducted on a Wikipedia graph, which is smaller than the real-world web
>
> In this paper we evaluate on the full Wikipedia (English version), which is still very large (over 5 million pages, ~300 million edges), being the first work to demonstrate a viable solution to navigation at this scale. Moreover, (i) nothing about our approach is specific or limited to Wikipedia. (ii) our proposed RFBC approach scales well: training and inference are local methods, i.e. only look at the immediate neighborhood of the current node, thus their wall-time does not grow significantly as the graph size increases (see response to reviewer J6Fp and Appendix E). Thus our method can potentially be applied to the entire web graph. However, there are significant practical obstacles to training on real-world web data (e.g. legal requirements concerning the rights to crawl/display/train on generic webpages), which put it out of scope for the current work.
>
>
> > I wonder if the authors have insights into why RL performs more poorly than the proposed RFBC (Table 2)?
>
> BC greatly outperforms RL in our setting because our navigation task is defined in such a way that we can generate virtually infinite expert trajectories ($O(N\times E_{out}^{T-1}$) trajectories), as opposed to the usual case in BC where the number of human annotations are limited. In many other settings the limited number of BC traces mean that RL performs well because it allows the agent to train on more environment data. But in our setting, with plentiful BC trajectories, the expert trajectory is a much stronger training signal than a sparse RL reward signal. This is one of the reasons why using random walk training is so effective: it gives us rich BC data with the same amount of environment interaction and exploration as RL. We have updated the paper to incorporate this point.
>
>
> > Two more related work discussion
>
> We thank the reviewer for pointing us to these relevant papers. We will add discussion to the final draft. In particular, LinkBERT is focusing on improving contextual representation using the graph structure, but doesn’t do any kind of navigation and it is not clear if such training will help. For the learning to retrieve paper, we are learning to navigate and tackle long-path navigation, as the reviewer points out.

---

> > ### Comment · Reviewer_L8iM · 2022-08-09
> > **Thank you for your response**
> >
> > Thank you so much for answering my questions and concerns, in particular, clarifying the main contribution of the paper. I have increased my rating.

---

> ### Author Response · Authors · 2022-08-08
> **Your thoughts on our response**
>
> Thanks again for the detailed feedback. As the discussion period is ending shortly, we wanted to check if you had a chance to look at our response. We have tried to address the comments from your initial review, but would be happy to discuss any points further.

---

### Official Review · Reviewer_ewnP · 2022-07-11

**Rating:** 5
**Confidence:** 4
**Soundness:** 2 fair
**Presentation:** 3 good
**Contribution:** 2 fair

**Summary:**

This paper focuses on the problem of web navigation. The authors proposed to train an agent which can navigate a large Wikipedia graph. They first generate sampled trajectories on the Wikipedia graph based on different methods (e.g. random forward trajectories, reverse trajectories, shortest paths). With these sampled trajectories, the behavioral cloning (i.e. supervised learning) method is adopted to train the model. The experiments show that the proposed method is not only able to efficiently navigate Wikipedia graph but also have competitive performance on the fact verification task.

**Questions:**

* If RL is adopted after behavioral cloning (i.e. RFBC+RL in Table 2), what's the performance?
* What are the training details, e.g. hardware, and training time?
* line 301, table 2 -> table 4?

**Limitations:**

The authors have discussed the limitation of this work.

**Strengths And Weaknesses:**

(1) Strengths
* The proposed method works well on navigation task and obtains competitive performance on the fact verification task.
* This paper is generally well written.

(2) Weaknesses
* Nogueira and Cho (2016)  is the most relevant work. The proposed method WebNav in Nogueira and Cho (2016) only gets 12.5% success rate, while the proposed method in this paper obtains more than 90% success rate in a similar setting. The reason for this large difference of performance is not explained or discussed in the experiments.

---

> ### Author Response · Authors · 2022-08-02
> **Response**
>
> We thank the reviewer for the thoughtful comments. We are glad that the reviewer found our proposed solution to be effective and the presentation to be of very high quality. We also would like to point the reviewer’s attention towards an additional task of information gathering for question answering on Natural Question (NQ) benchmark provided in the Appendix C.2 on which we obtain strong performance.
>
>
> > Difference of performance wrt to Nogueira & Cho (2016) not discussed.
>
> We apologize for not addressing this point satisfactorily. The dramatic performance gap stems from two factors: first, we carefully investigated, using the smaller 200k graph, effective architectures and training methods for navigation. Once we established a viable approach, we scaled up both the data and the model. For the former, our training graph was significantly larger than Nogueira & Cho (2016): 12M vs 38M nodes and 51.5M vs 380M edges. For the latter, we made use of contemporary architectures that scale effectively, i.e. BERT embeddings and Transformers, which are more powerful than the bag-of-words (BoW) and LSTM models used by Nogueira & Cho (2016). Collectively, these improvements deliver a dramatic difference in performance, with our model mostly solving the navigation task on the Wikipedia graph. We have revised the text to include this explanation.
>
>
> > If RL is adopted after behavioral cloning (i.e. RFBC+RL in Table 2), what's the performance?
>
> We did originally try adding RL after behavior cloning, but saw no statistically significant improvement over RFBC ($\sim1$ point difference in either direction). We added a new row for RFBC+RL to Table 2 and in Appendix C.1 we have added an example training curve to show that the accuracy fluctuates from the point where RFBC training left off.
>
>
> > What are the training details, e.g. hardware, and training time?
>
> For training we use 8 TPUv3 from Google Cloud for all our experiments. We pretrained for 100k steps for navigation which took $\sim9$ hrs ($\sim22$ hrs for the sentence search variant), followed by $\sim8$ hrs for downstream finetuning.
>
>
> > line 301, table 2 -> table 4?
>
> Thank you. We fixed the typos in the revised draft.

---

> ### Author Response · Authors · 2022-08-08
> **Your thoughts on our response**
>
> Thanks again for the detailed feedback. As the discussion period is ending shortly, we wanted to check if you had a chance to look at our response. We have tried to address the comments from your initial review, but would be happy to discuss any points further.

---

### Official Review · Reviewer_J6Fp · 2022-07-12

**Rating:** 6
**Confidence:** 4
**Soundness:** 3 good
**Presentation:** 2 fair
**Contribution:** 3 good

**Summary:**

In this paper, the authors study the web navigation problem. They propose behavioral cloning of the random trajectory method on Wikipedia graphs. They further apply their method to the FEVER verification task. Their experiments show that their method is effective in learning entity embeddings and a navigation policy. It also succeeds in a competitive performance in the verification task.

**Questions:**

Please refer to the major weaknesses mentioned above.

**Limitations:**

Yes, they mentioned two limitations of their work.

**Strengths And Weaknesses:**

Strengths:
1) The proposed method is more effective than the baseline methods and scalable to large Wikipedia graphs.
2) The extensive analysis of results and limitations of the study are presented.

Weaknesses:
Major:
1)  The proposed trajectories are very intuitive and lack novelty. They simply use random walks with the probabilities being decided based on neighboring vertices.
2) The proposed method is only compared with random and greedy methods, not other state-of-the-art methods. Hence, it is difficult to measure effectiveness.
3) A comprehensive efficiency analysis would be helpful. Comparison between the running times of the proposed methods and other methods would give a better understanding of the method.

Minor:
1) There are some typing issues. For example, on page 8, line 301, "Table 2" should be "Table 4".

---

> ### Author Response · Authors · 2022-08-02
> **Response**
>
> We thank the reviewer for taking the time to review our paper and recognizing that our proposed navigation method is intuitive, effective, and extensively analyzed. We also would like to point the reviewer’s attention towards an additional task of information gathering for question answering on Natural Question (NQ) benchmark provided in the Appendix C.2 on which we obtain strong performance.
>
> >  The proposed trajectories are very intuitive and lack novelty.
>
> We consider the simplicity of our approach to be a virtue. Instead of using more elaborate methods for generating good paths like shortest path, our novelty lies in showing the random walks suffice to train an effective navigation policy.  Along with empirical evidence, we add a more theoretical justification in the Appendix D from a variational inference perspective. We also note that training using random walks is akin to pretraining in Vision and NLP domains using random masking and the simplicity of the approach also allow it to scale.
>
>
> > The proposed method is only compared with random and greedy methods, not other state-of-the-art methods. Hence, it is difficult to measure effectiveness.
>
> We respectfully disagree: we show comparisons to navigation with Random, Greedy, DFS, and RL as listed in Table 2. We also showed two variants of DFS: Greedy DFS and Random DFS which pick the next node to visit using the popular heuristics of randomly or greedily.  Also we would like to point out that the problem is understudied -- we are not aware of any other sophisticated baseline to compare to. However, we are happy to compare to any other baselines that the reviewer might  suggest. Furthermore, on the downstream FEVER task and NQ task we do compare to leaderboard SOTA Stammbach et al 2021 and Qu et al 2021 respectively.
>
>
> > A comprehensive efficiency analysis would be helpful. Comparison between the running times of the proposed methods and other methods would give a better understanding of the method.
>
> The proposed RFBC training and inference are local methods (i.e only look at the immediate neighborhood) whose cost is nearly constant in the graph size (i.e. there is no search procedure at inference time). The runtime of our method is based on the length of the trajectory T and the average number of links on a page. While in theory, this would grow (very) slowly with increasing graph size, in practice, the number of outbound links in a paragraph is upper-bounded and does not really grow with graph size. Our observed training time difference between the 200k node graph and the full 38M node graph is only around 1.5x despite an almost 200-fold increase in graph size. We summarize the asymptotic running times of the proposed methods and other methods in the following table (also added to the Appendix E), where we see that RFBC scales more gracefully than other effective approaches.
>
> | Method | Cost (Big-O) | Accuracy (5 step 200k graph) | Scalable |
> | ----------- | ----------- | ----------- | ----------- |
> | RFBC (Ours) | $O(E_\text{out} T)$ | 85.3% | Yes |
> | Backwards BC | $O(E_\text{out} T)$ | 2.3% | Yes |
> | Shortest path BC | $O(E \log V + E_\text{out} T)$ | 86.7% | No |
> | RL | $O(E_\text{out} T)$ | 41.4% | Yes |
> | Random | $O(T)$ | 12.3% | Yes |
> | Greedy | $O(E_\text{out} T)$ | 19.7% | Yes |
> |Random DFS | $O(T)$ | 10.0% | Yes |
> | Greedy DFS | $O(E_\text{out} T)$ | 31.1% | Yes |
>
>   $V$: nodes in the graph [large -- $10^7$],
>   $E$: total edges in the graph  [very large -- $10^8-10^9$],
>   $E_{out}$: average out degree of node (\# of actions)  [small -- $10^1$],
>   $T$: length of trajectory, i.e. the maximum steps agents can run. [small -- $10^1$],
>
>
> > There are some typing issues. For example, on page 8, line 301, "Table 2" should be "Table 4".
>
> Thank you. We fixed the typos in the revised draft.

---

> ### Author Response · Authors · 2022-08-08
> **Your thoughts on our response**
>
> Thanks again for the detailed feedback. As the discussion period is ending shortly, we wanted to check if you had a chance to look at our response. We have tried to address the comments from your initial review, but would be happy to discuss any points further.

---

### Meta-Review · Area_Chair_hn5z · 2022-08-28

**Recommendation:** Accept
**Confidence:** Certain

**Metareview:**

This paper aims to use the random walk to learn an effective link selection policy on a graph version of Wikipedia. They navigate graph-structured web data to find a target by hyperlinks within articles. To effectively navigate on the web, they first construct Wikipedia as a graph where nodes represent a web page and edges denote the hyperlinks in the current page and then conduct several strategies for sampling random trajectories from the graph to find the path from the start node to target node.

Overall, this paper is interesting that building Wikipedia as a graph to navigate the target. However, the novelty of this paper is not enough. Multi-hop reasoning in the knowledge graph has achieved significant progress. The main idea of this paper is similar to multi-hop reasoning. Besides, the authors only use the existing path-finding method, such as random walk, to navigate the target. Although the authors claim that difference between their navigation and knowledge graph navigation, a better path-finding method satisfied Wikipedia navigation should be explored.

Since the constructed Wikipedia graph contains natural language text, the authors
use a Transformer to learn representations for nodes and edges. Benefiting from semantic content, the learned representations can enhance the performance of downstream tasks. Such a method is a common technique applied in text graphs where nodes consist of text information. The novel aspects only lie in the navigation policy network, which computes possible actions and defines the probability for action selection. Instead of using pre-training embedding for action embeddings, the authors utilize learnable embeddings for actions.

The authors apply the proposed navigation method to Wikipedia for the fact verification task and achieve a significant boost when integrated into a simple TF-IDF scheme.
Results in small graph navigation show that the proposed method accomplishes an outstanding performance in terms of success rate. Besides, results demonstrate that the learned embeddings perform better than fixed embedding from pre-trained language models. For example, the navigation policy network with a feed-forward layer performs best in most cases. The experimental results are consistent with the assumption.



**Award:**

No

---

### Meta-Review · Area_Chair_iuX4 · 2022-09-14

**Recommendation:** Accept
**Confidence:** Certain

**Metareview:**

I concur with the decision of the meta reviewer to accept the paper. While the technical novelty may be limited, the excellent empirical performance of the overall system of many components is an interesting point to share with the community.

**Award:**

No

---

### Decision · Program_Chairs · 2022-09-14

Accept